# Extracting Meaningful Attention on Source Code: An Empirical Study of Developer and Neural Model Code Exploration

## Abstract

The high effectiveness of neural models of code, such as OpenAI Codex and AlphaCode, suggests coding capabilities of models that are at least comparable to those of humans. However, previous work has only used these models for their raw completion, ignoring how the model reasoning, in the form of attention weights, can be used for other downstream tasks. Disregarding the attention weights means discarding a considerable portion of what those models compute when queried. To profit more from the knowledge embedded in these large pre-trained models, this work compares multiple approaches to post-process these valuable attention weights for supporting code exploration. Specifically, we compare to which extent the transformed attention signal of CodeGen, a large and publicly available pre-trained neural model, agrees with how developers look at and explore code when each answering the same sense-making questions about code. At the core of our experimental evaluation, we collect, manually annotate, and open-source a novel eye-tracking dataset comprising 25 developers answering sense-making questions on code over 92 sessions. We empirically evaluate five attention-agnostic heuristics and ten attention-based post processing approaches of the attention signal against our ground truth of developers exploring code, including the novel concept of *follow-up attention* which exhibits the highest agreement. Beyond the dataset contribution and the empirical study, we also introduce a novel practical application of the attention signal of pre-trained models with completely analytical solutions, going beyond how neural models' attention mechanisms have traditionally been used.

## 1 Introduction

Recent large neural source code models such as Codex (Chen et al., 2021), CodeGen (Nijkamp et al., 2022) and AlphaCode (Li et al., 2022) are remarkably effective at program synthesis and competitive programming tasks respectively. Yet our understanding of why they produce a particular solution is limited. In practice, the models are mostly used for their prediction alone, i.e., as generative models, and the way they reason about code internally largely remains untapped.

These models are often based on the attention mechanism (Bahdanau et al., 2016), a key component of the transformer architecture (Vaswani et al., 2017). Besides providing substantial performance benefit, attention weights have been used to provide interpretability of neural models (Lin et al., 2017; Vashishth et al., 2019; Paltenghi & Pradel, 2021). In particular, Wan et al. (2022) and Vig & Belinkov (2019) have shown how the attention weights contain important syntactic information on both the Abstract Syntax Tree (AST) of source code and Part of Speech (POS) tags in natural language. Moreover, Wan et al. (2022) showed how using attention weights to infer the distance between two tokens outperformed techniques using hidden representations. In a similar direction, Zhang et al. (2022) has shown how a novel graph representation of source code derived solely from attention weights achieved comparable performance on the VarMisuse dataset (Allamanis et al., 2018) to that of a hand-crafted graph representation based on control flow and data flow (Hellendoorn et al., 2020).

The work cited above suggests the attention mechanism reflects or encodes objective properties of the source code processed by the model. We argue, that just as software developers consider different locations in the code individually and follow precise connections between them, so the self-attention of transformers connects and creates information flow between similar and linked code locations. If those relations are indeed comparable, this raises the possibility: *can the knowledge about source code conveyed by the attention weights of neural models be leveraged to support human code exploration?*

There are datasets tracking developers' visual attention while looking at code, but they do not seem suitable to this task. The largest ones either put the developers in an unnatural (and thus possibly biasing) environment where most of the vision is blurred (Paltenghi & Pradel, 2021), or they contain few and very specific code comprehension tasks (Bednarik et al., 2020) on code snippets too short to exhibit any interesting code navigation pattern.

**This work.** To address these limitations and stimulate developers and the neural model to not only glance at code, but also deeply reason about it, we prepare an ad-hoc code understanding assignment called *sense-making task*. This involves questions on code including mental code execution, side effects detection, algorithmic complexity, and deadlock detection. We collect an eye-tracking dataset of 92 valid sessions with developers. The sense-making task is additionally designed to be machine friendly with a specific prompt to trigger a completion from the model which hopefully stimulates the reasoning of the model. Then we query CodeGen on the same sense-making task and compare its *attention signal*[1] to the attention of developers. They turn out to be positively correlated (r=+0.23), motivating the use of raw and processed versions of the attention signal for code exploration. To that end, we experimentally evaluate how well existing and novel attention post-processing methods align with the code exploration patterns derived from the chronological sequence of eye-fixation events of our dataset. To the best of our knowledge, this work is the first to investigate the attention signal of these pre-trained models to support code exploration, a specific code-related task.

We empirically demonstrate that post-processing methods based on the attention signal can be well aligned with the way developers explore code. In particular, using the novel concept of *follow-up attention* we achieve the highest overlap with the top-3 developers' ground truth on which line to explore next.

**Main contributions.** Our key contributions are:

- A novel dataset of eye tracking data, comprising 92 visual attention sessions of 25 developers engaged in sense-making tasks while using a common code editor with code written in three popular programming languages (Python, C++ and C#).

- The first experimental comparison of both effectiveness and visual attention of GPT-like models and developers when reasoning on sense-making questions.

- Demonstrating a connection between the neural attention signal and the temporal sequence of location shifts regarding developer focus.

- The analytical formula for *follow-up attention*, a novel post-processing approach derived solely from the attention signal, which aligns well with the developer interaction of which line to look at next when exploring code.

- An empirical evaluation comprising ten post-processing approaches of the attention signal, five heuristics, and an ablation study of the follow-up attention against the collected ground truth of developers exploring code.

## 2 RELATION TO EXISTING WORK

**Attention as explanation.** Previous work (Jain & Wallace, 2019) on studying attention weights of recurrent neural models has found that the attention weights do not always agree with other explanation methods and that alternative weights can be adversarially constructed while still preserving the same model prediction. However, in response, a successive study (Wiegreffe & Pinter, 2019) showed how the alternative attention weights can be constructed only per a single instance prediction, whereas obtaining a model which is consistently wrong in its explanations is very unlikely

---

[1] Attention signal refers to the attention weights produced during a forward pass by the transformer blocks.

to happen. On the same line, Tutek & Šnajder (2020) has proposed four regularization methods to mitigate the adversarial exploitation of attention weights for recurrent models, including the use of residual connections which are natively embedded into transformers (Vaswani et al., 2017), the building blocks of GPT-like models Radford et al. studied in this work.

**Alternatives to attention signal.** There are observable signals other than attention that might capture the concept of relevance, such as gradients-based or layer-wise relevance propagation Montavon et al. (2019). In particular, Yuan et al. (2021) and Chefer et al. (2021) are promising examples of approaches based, partially or entirely, on gradient information. However, we focus on approaches using only the attention signal for two reasons: (1) almost all state-of-the-art models of code are based on the transformer block, and the attention mechanism is ultimately its fundamental component (Vaswani et al., 2017), so we expect the corresponding attention weights to carry directly meaningful information about the decision process of these transformer-based models; (2) unlike the gradients, the attention weights can be extracted almost for free during generation with little runtime overhead, a crucial condition in many practical scenarios.

**Attention studies of neural models of code.** Paltenghi & Pradel (2021) have compared the attention weights of neural models of code and developers' visual attention when performing a code summarization task, and found a strong positive correlation on the *copy attention* mechanism for an instance of a pointer network (Vinyals et al., 2015). Wan et al. (2022) and Zhang et al. (2022) have then shown how the attention weights of pre-trained models on source code capture important properties of the abstract syntax tree of the program. However, none of them considered the use of the attention signal for a code related task, such as code exploration. Moreover, they are limited to relatively small self-attention transformer models, whereas we study the attention of CodeGen Nijkamp et al. (2022), a large generative model using masked self-attention.

We point the interested reader to Appendix B for a additional connections with existing literature.

## 3 CODE UNDERSTANDING TASK: SENSE-MAKING QUESTIONS

To study developers' and models' attention, we prepare a code understanding task called the *sense-making task*. One sense-making task is contained in a single source code file $p$ composed of four sections: (1) a brief description of the context of the main code snippet (e.g. *The following code reasons about triangles in the geometrical sense.*), (2) the main code snippet, either sourced on the internet or written from scratch by the authors, (3) a sense-making question to stimulate the reasoning (i.e., `Question:`), and (4) a final prompt to trigger the model's answer (i.e., `Answer:`).

```
1   # ***********************************************************************
2
3   # The following code reasons about triangles in the geometrical sense.
4
5
6   class point:
7       def __init__(self, x, y):
8           self.x = x
9           self.y = y
10
11  def square(x):
12      return x * x
13
14  def order(a, b, c):
15      copy = [a, b, c]
16      copy.sort()
17      return copy[0], copy[1], copy[2]

56  p1 = point(0, 0)
57  p2 = point(1, 1)
58  p3 = point(1, 2)
59  classifyTriangle(p1, p2, p3)
60
61
62  # Question: What could happen if the call to `order()` were omitted from
63
64  # `classifyTriangle`?
65
66  # Answer:_
```

Figure 1: Example of sense-making task with code and question to be answered in the bottom comment.

Note that all the sections except the main snippet are in the form of code comments. The questions include a variety of topics such as mental code execution, side effects detection, algorithmic complexity, and deadlock detection; Figure 1 shows an example task, whereas the full list of questions can be seen in the Appendix A. We prepare five main snippets and create three unique questions for each of them, then we translate the same task in three programming languages: Python, C++ and C#. In total, we have 45 unique tasks.

**Neural Model's Task**. We feed the entire source file corresponding to a single task as input of the generative model, and query it with temperature $t = 0.2$ to generate three different answers in the form of text completion. We fix the maximum number of generated tokens to 100. The model used is CodeGen (Nijkamp et al., 2022), in its largest language agnostic variant [2].

---

[2] We use `CodeGen-16-multi` from `https://github.com/salesforce/codegen`.

**Developers' Task**. We recruit 25 software developers via direct contacts at a large software company, without any constraint on prior programming experience. We track the eye gaze of each participant during a session of a maximum of 45 minutes while they answer as many questions as possible, typically three or four. We ensure that they see each main code snippet only once to avoid bias in answering a question on a snippet they have already explored in a previous task. To ensure consistent data collection, the eye tracking setup is calibrated at the beginning of each task.

## 4 PROBLEM FORMULATION: ATTENTION

The large majority of large language models is based on generative pre-trained transformers (GPT) (Radford et al.). This is true also for models of code with notable examples such as Codex (Chen et al., 2021), CodeGen (Nijkamp et al., 2022), and AlphaCode (Li et al., 2022). GPT-like models such as these are comprised only of decoder layers rather than of both encoder and decoder layers, and use *masked* self-attention to prevent a token from attending tokens which come after it. In GPT-like models the self-attention given by token $i$ to the other tokens in the sequence can be represented by a vector: $\boldsymbol{a}_i = (a_{i,1}, a_{i,2}, ..., a_{i,i}, 0_{i,i+1}, ..., 0_{i,n})$ where $n$ is the maximum allowed sequence length. Note that token $i$ has one such vector for each combination of transformer layer and attention head. For each layer and head, the attention weights $a_{i,j}$ form a triangular matrix.

### 4.1 VIEWS OF ATTENTION

Given a GPT-like neural model $f$ and and a sense-making task $p$, whenever the model ingests $p$, its tokenizer splits it in $n$ prompt tokens $p = t_1, ..., t_n$ and it outputs both the original input sequence followed by newly generated tokens and the attention tensor corresponding to the masked self-attention weights:

$$f(p) = (t_1, ..., t_n, t_{n+1}, ..., t_{n+m}, \mathbf{A}) \tag{1}$$

where $m$ is the number of generated tokens and $\mathbf{A}$ is the 4-dimensional attention tensor of shape $(l, h, n + m, n + m)$ where $l$ is the number of layers, $h$ is the number of attention heads, $n$ is the number of tokens in the prompt. Note that unlike the predicted tokens, which are randomly sampled with a temperature parameter, the attention weights stay the same across runs provided that the input prompt remains the same. In particular, when comparing developers' and the model's attention, we focus on studying the attention weights referring to the prompt tokens only, thus the dimension of $\mathbf{A}_{prompt}$ are $(l, h, n, n)$ even if some post-processing approach may use the entire tensor.

Each post-processing approach of the attention signal can be represented by an *extraction function $g$* which transforms the attention tensor $\mathbf{A}$ into one of two views: *visual attention* or *interaction matrix*.

**Visual Attention.** The visual attention is a static view which tells us what part of the input is important for the model when solving the sense-making task. We define the *visual attention* of a model or a developer as a vector $\boldsymbol{a} = (a_1, ..., a_c)$ over the single characters of the prompt, where each $a_i$ intuitively tells us the how much "attention" was given to that the $i$-th character when solving the task. We represent the corresponding extraction function as $g_{viz}(\mathbf{A})$, which is a function that takes as input the attention tensor $\mathbf{A}$ and returns the visual attention $\boldsymbol{a}$.

**Interaction Matrix.** The interaction matrix is a dynamic view which tells us, given a position in the prompt, which other position of the prompt should likely be attended soon after the current position by the model. Intuitively, the interaction matrix can be directly used to support the code navigation of a developer since it could recommend to the developer the likely next interesting piece of code at any point in time. We define the interaction matrix $\boldsymbol{S}$ as a right stochastic matrix with size $n \times p$ where $n$ is the number of tokens in the prompt and $p$ is the number of admissible target positions in the prompt. We define two interaction matrices depending on the granularity of the target position $p$: (1) *token-level*, where $\boldsymbol{S}$ has size $n \times n$ where $n$ is the number of tokens in the prompt; (2) *line-level*, where $\boldsymbol{S}$ has size $n \times n_l$ where $n_l$ is the number of lines in the prompt. We represent the corresponding extraction function as $g_{inter}(\mathbf{A})$, which is a function that takes as input the attention tensor $\mathbf{A}$ and returns the interaction matrix $\boldsymbol{S}$. In particular, we distinguish between the two granularity levels using two functions $g_{token}$ and $g_{line}$ leading to $\boldsymbol{S}_{token}$ and $\boldsymbol{S}_{line}$ respectively.

## 5 METHODOLOGY: EXTRACTION FUNCTIONS

For the empirical evaluation, we investigate two concrete extraction functions for the the visual attention and four for the interaction matrix.

### 5.1 EXTRACTION FUNCTIONS FOR VISUAL ATTENTION

To condense the attention tensor $\mathbf{A}$ to a vector $\boldsymbol{a}$, we introduce two straightforward approaches: *attention mean* and *attention max*.

**Attention mean.** It simply takes the average attention weight for each position along the two dimensions of different layers $l$ and attention heads $h$ to compensate the extreme values with each other and extract a smoother signal. This step, outputs a matrix $\boldsymbol{A}$ with shape $(n, n)$ where each element $\boldsymbol{A}_{i,j}$ is the average attention weight of the $i$-th token in the prompt to the $j$-th token in the prompt. Note that it is a lower triangular matrix due to the masked nature of the self-attention. Then, since the positions above the main diagonal are all zeros, we compute the mean of each column excluding the zeros to avoid penalizing more recent tokens with less followers. Since each column can be seen as the attention given to $j$-th token by its "followers" we call the step *mean of followers*, which produces a token-level visual attention $\boldsymbol{a}$ vector. Finally, to convert it to a character level vector, we divide the attention weight on a single token in *equal shares* among all its characters.

**Attention max.** This approach differs from the previous one in the way it condenses layers and heads replacing the mean with the max function to favour the extreme positive signals appearing only in one or few layers and heads, the rest is unchanged.

### 5.2 EXTRACTION FUNCTIONS FOR INTERACTION MATRIX

We study four approaches: *max*, *mean*, *rollout* and *follow-up attention*. Apart from the rollout attention, which has been introduced by Abnar & Zuidema (2020), the other three are either inspired by the work of Paltenghi & Pradel (2021) or a novel contribution of this work, such as the follow-up attention.

**Attention mean.** It computes the mean among all the $L$ layers and $H$ attention heads: $g_{mean} = \frac{1}{l \cdot h} \sum_{l=1}^{L} \sum_{h=1}^{H} \mathbf{A}_{l,h}$.

**Attention max.** It computes the max among all the $L$ layers and $H$ attention heads: $g_{max} = \max_{l=1}^{L} \max_{h=1}^{H} \mathbf{A}_{l,h}$.

**Rollout attention.** It propagates the information contained in the attention weight layer by layer from input to output, by multiplying the attention weights along a multiple paths starting and ending to the same input-output pair. Since it does not model the attention head dimension, we simply condense the that dimension via a simple sum. Moreover, it does not prescribe which attention layer should be used in the end, thus we sum the newly computed rollout values of all the layers.

**Follow-up attention.** It is a novel approach that we introduce in this work. Similarly to the rollout attention, we aggregate the attention weights over the $h$ attention heads by summing the weights of $\mathbf{A}$ along the attention head

---

**Algorithm 1** Follow-up Attention

1: **Input: A** {Dim: $(L, H, n + m, n + m)$}
2: **Output: $\boldsymbol{S}$**
3: $consec\_pairs \leftarrow \emptyset$
4: $\mathbf{L} \leftarrow \sum_{h=1}^{H} \mathbf{A}_h$ {Layer-wise attention}
5: **for** $z \leftarrow 1$ to $L - 1$ **do**
6: $\quad \boldsymbol{S}^{(z)} \leftarrow \emptyset$
7: $\quad$ **for** $i \leftarrow 1$ to $n$ **do**
8: $\quad\quad$ **for** $j \leftarrow 1$ to $n$ **do**
9: $\quad\quad\quad s \leftarrow n + m - \max(i, j)$
10: $\quad\quad\quad \boldsymbol{f}_i^{(z)} \leftarrow \mathbf{L}_{z,s:,i}$
11: $\quad\quad\quad \boldsymbol{f}_j^{(z+1)} \leftarrow \mathbf{L}_{z+1,s:,j}$
12: $\quad\quad\quad \boldsymbol{S}_{i,j}^{(z)} \leftarrow \frac{\boldsymbol{f}_i^{(z)} \cdot \boldsymbol{f}_j^{(z+1)}}{||\boldsymbol{f}_i^{(z)}|| \cdot ||\boldsymbol{f}_j^{(z+1)}||}$
13: $\quad\quad$ **end for**
14: $\quad$ **end for**
15: $\quad consec\_pairs \leftarrow consec\_pairs \cup \boldsymbol{S}^{(z)}$
16: **end for**
17: $\boldsymbol{S} \leftarrow \sum_{z=1}^{L-1} \boldsymbol{S}^{(z)}$

---

dimension and obtaining the *layer-wise attention* $\mathbf{L} = \sum_{h=1}^{H} \mathbf{A}_h$. We leave the exploration of head-level aggregation strategies to future work, and rather focus on modelling the information flow between subsequent layers. We refer to this condensed 3-dimensional tensor as *layer-wise attention* represented by $\mathbf{L}$. The follow-up attention explicitly models the temporal relationship between the

attention weights computed at different layers, since the attention weights in the later layers depend on the earlier ones. The intuition is that the layer-after-layer transformation reflects the way in which the models explores code through time, similarly to multiple successive fixations of a developer when navigating and exploring source code. Instead of looking at how the a token gives attention to other tokens in the same layer, the follow-up attention adopts a differential approach which compares the attention *received* by token $i$ at layer $z$ with the attention *received* by token $j$ at layer $z - 1$. To represent this received attention we define the *follower score* $\boldsymbol{f}_i^{(z)}$ of token $i$ at layer $z$, as the vector of the attention quota that each other token gives to token $i$ at the same layer. Note that, similarly to the self-attention vector, the follower score is also a vector of real numbers and it has the same length corresponding to the input sequence length, thus representing a complementary viewpoint. To realize agreement between follower scores at two consecutive layers, we use the cosine similarity as a soft version of the intersection between the set of followers of the two tokens. Then we compute the follow-up attention for each ordered pair of tokens ($i \rightarrow j$ for $i$ prompt or generated token) and for each pair of consecutive layers, and condense all layer pairs into a single matrix via sum. We aggregate attention over multiple layers, since Brunner et al. (2020) have empirically shown how token identifiability is retained over layers, thus a generic embedding at position $\boldsymbol{e}_i$ in any layer $l$ is traceable to the input embedding $\boldsymbol{x}_i$ in the input sequence. Their finding can be partially explained by the residual connections of transformers, which were studied by (Abnar & Zuidema, 2020). Algorithm 1 summarizes the overall algorithm. An optimized vectorized version is available here [3].

## 6 EXPERIMENTAL EVALUATION

In this section, we compare the visual attention and interaction matrix extracted from the attention tensor of neural models against the ground truth computed from the developers. All our code is publicly available at `https://anonymous.4open.science/r/attentionStudyArtifacts-C752/` and the dataset is available here [4]

**Eye Tracking Dataset.** We derive the ground truth from our eye tracking dataset comprising 25 participants over 92 valid sessions (see Appendix E for details). Each session consists of a sequence of eye fixation events $evt_{eye}$, each represented as a tuple $(t, x, y, d)$ where $t$ is the timestamp in milliseconds, $x$ and $y$ are the coordinates of the fixation point in pixels and $d$ is the duration of the fixation in milliseconds. Each session is recorded in Visual Studio Code[5] to have a natural coding environment. Based on the size of the parafoveal region (Schotter et al., 2012), each eye fixation event is converted to column and line coordinates: $evt_{eye}^{(char)} = (t, c, l, d)$ where $c$ is the column and $l$ is the line of the original source file (more detail in Appendix D).

**Ground truth visual attention.** Here we borrow from the concept of *human attention* proposed by Paltenghi & Pradel (2021) and define the analogous *developer attention* as the total time that a specific char was visible to the participant: $\boldsymbol{d} = (d_1, ..., d_c)$ where $c$ is the number of characters in the prompt and $d_i$ is the total time that the $i$-th character was visible to the participant according to the eye tracking data. In difference to Paltenghi & Pradel (2021), we consider the char-level instead of the token level, which is the granularity in many contexts, and in particular for eye tracking data.

**Ground truth interaction matrix.** From each developer session we derive a ground truth interaction matrix $\boldsymbol{S}$. For a fair comparisons of neural models with developers, we take into account the tokenization used by the neural model, namely we use the CodeGen tokenizer [6] which is based on byte-level byte-pair-encoding (Sennrich et al., 2016).

To convert char-level events into token level ones, for each timestamp, if at least one character of a given token is visible, then the token is considered visible as well and we count the corresponding event $evt_{eye}^{(token)} = (t, i, d)$ where $t$ is the timestamp, $i$ is the token index and $d$ is the event duration. Based on the pairs of events involving token $i$ and token $j$, we quantify how likely it is that the developer looks at token $j$ after having looked at token $i$.

---

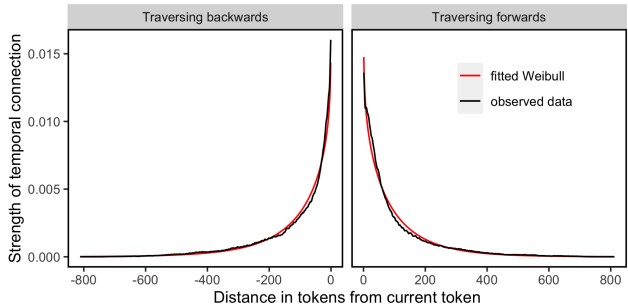

Figure 3: The strength of the connection $S_{i,j}$ depends significantly on the difference $i - j$. As detailed in Appendix F, both cases $i > j$ and $j > i$ can be well modelled using a Weibull distribution.

Intuitively, we want to have stronger connection when a fixation on token $i$ is shortly followed by a fixation on token $j$, and if this second fixation has a significant duration. Thus, we define the *strength of the temporal connection* between token $i$ and token $j$ as:

Figure 2: Example of two events where the yellow area corresponds to their contribution to the connection strength between from token $i$ to token $j$.

$$S_{i,j} = \sum_{evt_i, evt_j \in P_{i \to j}} \int_{t_j}^{t_j + d_j} e^{-\alpha(t - (t_i + d_i))} dt \quad (2)$$

where $P_{i \to j}$ is the set with all the pairs of events where token $i$ is seen before token $j$, and $\alpha$ is a parameter that controls the decay of the connection the more the two events are far apart in time. For our experiments we empirically set $\alpha = 0.1$, accounting for observed behaviour where developers often spend several seconds scrolling and presumably shallow searching the code. In Figure 2 we show an example of the integral connecting two consecutive events.

We noticed a strong *neighboring effect* across the whole dataset, where the connection between closer tokens tends to be relatively stronger irrespective of context and content (see Figure 3 and Appendix F). To extract relevance beyond mere closeness, we normalize each row of the interaction matrix $\boldsymbol{S}$ by dividing by the average empirical ground truth distribution where the probability to go to a token constantly decreases the further away the target token is.

**Agreement Metrics.** To measure the agreement between the visual attention of developers and the neural model, we regard them as vectors with meaningful ordinal content, and compute their *Spearman rank correlation coefficient* (Spearman, 1987). In contrast, we need to compute the agreement between the interaction matrices row by row, using either the Spearman rank correlation coefficient or the top-3 overlap, defined as the number of top-3 target positions shared between the ground truth and the model-derived interaction matrix. We focus on the line-level interaction matrix $\boldsymbol{S}_{line}$, since we argue that the line-level is perhaps the most useful granularity from an hypothetical user perspective. To obtain $\boldsymbol{S}_{line}$ we simply sum the probabilities to go to tokens belonging to the same line. Moreover, to balance the fact that some potential starting tokens might be rarely (or only very transiently) looked at by the developers, we weight each comparison based on the total number of seconds spent by the developer on the corresponding starting token. We also fix a maximum for this weight to 10 sec to prevent long-observed tokens dominating the comparison. We only ever directly compare tasks related to the same code snippet, and when we have more than one developer ground truth on the same task, we consider all the possible developer-model combinations independently.

## 6.1 DEVELOPERS VS NEURAL MODELS

Our initial investigation considers three neural models: CodeGen [7], GPT-J [8] and InCoder Fried et al. (2022). For the in-depth study of the interaction matrix we focus on the more recent and more performant CodeGen model.

---

[7]https://huggingface.co/Salesforce/codegen-16B-multi
[8]https://huggingface.co/EleutherAI/gpt-j-6B

**Answers correctness.** We manually grade developers and neural model answers as either correct, partially correct or wrong, with high inter-rater reliability indicating a reproducible annotation process (see Appendix C). The developer's answers are correct or partially correct almost 70% of cases, while the neural models show a promising and non-trivial performance (Figure 4, Appendix G). We note the possibility that were the neural models given a more sophisticated prompt than ours, i.e., `#Answer:`, their performance might have been even stronger, considering a multitude of results in prompt engineering research (Prenner & Robbes, 2021; Shrivastava et al., 2022; Reynolds & McDonell, 2021; Bareiß et al., 2022).

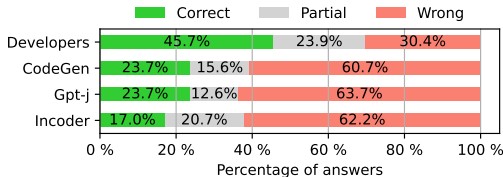

Figure 4: Percentage of correct, wrong and partially correct answer for developers and model.

**Agreement on visual attention.** In Figure 5, we report the Spearman rank correlation Spearman (1987) between the developer attention vector and the model attention vector, for completeness we also report the comparisons among developers. We assess the difference among programming languages with the Mann Whitney U Test (Mann & Whitney, 1947) applied to the distributions of Spearman coefficients, and we could only spot a significant difference between developers and models agreement on Python and C# code (p-value $< 0.05$), with a higher agreement obtained on C# (more in the Appendix H). This exceeds the agreement observed in previous work Paltenghi & Pradel (2021), which we hypothesize to be due to a combination of us using a more advanced model Ahmad et al. (2020) and a more natural data collection setup (eye tracking vs a deblurring interface).

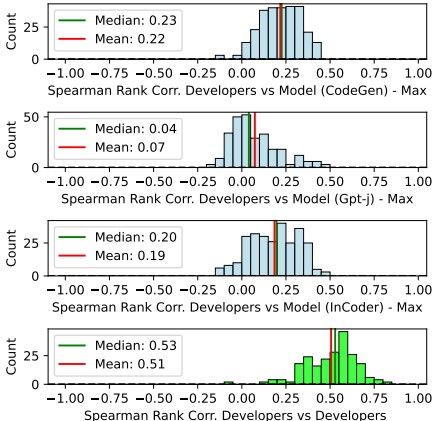

Figure 5: Agreement between developers and models' visual attention (extraction function: max).

Considering CodeGen's slightly better performance and significantly higher developer agreement, we restrict our subsequent investigation to this model.

**Agreement on interaction matrix.** We run the different extraction functions on the model attention signal to obtain interaction matrices $S_{line}$, which we compare to developer derived ground truth.

We distinguish between: (1) *attention-based code traversal* predictions, which are those introduced in Section 5.2, and (2) *attention-agnostic code traversal* predictions. The attention-based methods comprise raw attention in first and last layer, max, and mean, with their respective symmetric versions where the triangular matrix is mirrored and added to replace the zero values, the rollout and follow-up attention. The attention-agnostic methods comprise: *copy-cat* recommending all the positions containing the same starting token, *uniform* recommending all the position preceding the current token, and *position* recommending the neighboring positions of the current token with a Gaussian distribution centered on the current token.

We find that attention-based methods do carry predictive power, and in particular that *follow-up attention performs best* among all methods for both Spearman rank and top-3 overlap (Figure 6). We note that a purely position-based approach performs better than the copy-cat method despite being completely content-agnostic. We attribute this to developers' tendency to often read source code in (piecewise) linear order as described by (Blascheck & Sharif, 2019). Regarding raw attention, Zhang et al. (2022) demonstrated deeper semantic information being concentrated in later layers. Yet for both the triangular and symmetric versions respectively, higher levels appear inferior at predicting

eye-movement to earlier levels, possibly because such deeper semantic information is not directly apparent to developers at a glance.

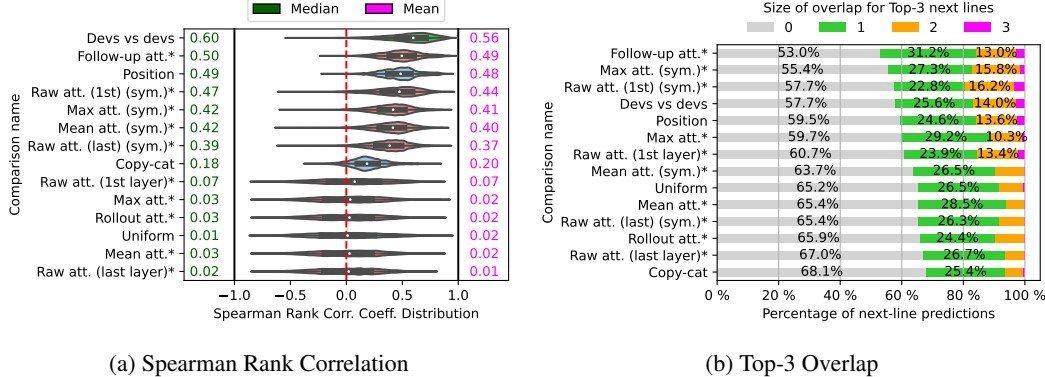

(a) Spearman Rank Correlation        (b) Top-3 Overlap

Figure 6: Agreement between developers and CodeGen interaction matrix ordered from weak (top lines) to strong (bottom lines). Attention-based methods are marked with *.

**Ablation study of follow-up attention.** We investigate two key design choices for follow-up attention: (1) the selection of layers to use, (2) the number of generated tokens, i.e., observers (Figure 7) There is significant agreement with the ground truth even for smaller numbers of layers, particularly the very first one. This suggests little processing, and maybe even only the token embedding, might be needed to extract valuable information. Additionally, a higher number of observers of the follow-up has a positive impact on the agreement of the follow-up attention with the ground truth.

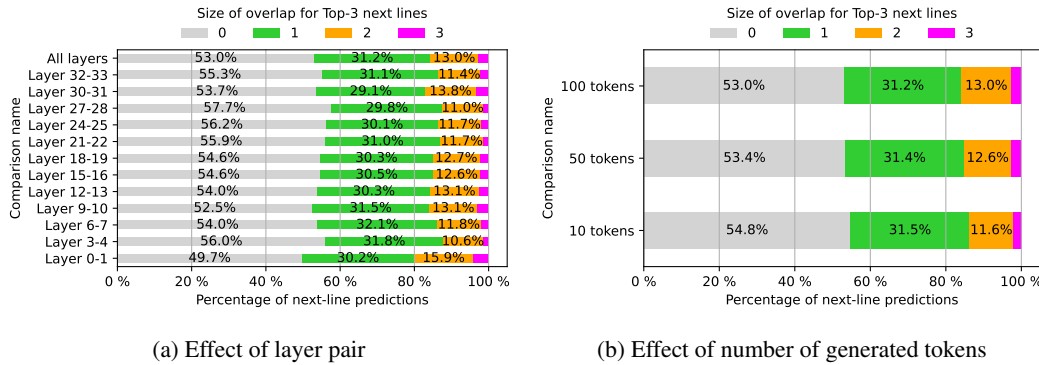

(a) Effect of layer pair        (b) Effect of number of generated tokens

Figure 7: Ablation study of Follow-up Attention. Left: Restriction to few layers still retains significant predictive power. Right: Generating more tokens improves the prediction.

# 7 CONCLUSION

We presented and shared a novel dataset of eye tracking data, comprising 92 visual attention sessions of 25 developers when answering sense-making questions in three popular programming languages (Python, C++ and C#). We confirmed that neural models provide promising, but less accurate answers than developers to these questions, while paying attention to similar parts of the code. We formalized a new code exploration task of predicting developer code traversal and confirmed the attention signal's relevance for this task by evaluating multiple processing approaches. Besides existing post-processing, we contributed the concept of follow-up attention, which shows the best agreement with the developer attention data.

## 8 ETHICS STATEMENT

The study involved human subjects who decided to participate voluntarily. Each of them gave explicit consent to use their data for scientific publication. Each eye-tracking session was limited to 45 minutes and included breaks to limit the cognitive stress on the subject. All the shared data have been anonymized for privacy by removing any reference to personal details of the participants, such as names. The eye-tracking data collection was carried out in an ergonomic setup with a non-invasive method, but with an eye-tracking camera pointing to the face of the participant. Each participant received a 10 USD company cafeteria food coupon for participating in the study. The authors have no conflict of interest to report.

## 9 REPRODUCIBILITY STATEMENT

All our code is publicly available at the following repository: `https://anonymous.4open.science/r/attentionStudyArtifacts-C752/`.

The eye tracking dataset is available both in its raw version with all the eye tracking sessions `https://figshare.com/s/d56af6f915bc29e6f620` and also in its processed version with interaction matrices computed as described in the main text `https://figshare.com/s/c20e04362097125fd27e`. The results of the empirical comparisons are also present in the last mentioned dataset. The repository contains a `Readme` and Python notebook to reproduce the main figures of the paper starting from the shared data.

In addition, since all the code used is publicly available, including the snippets used in the study and the code to process the eye tracking data, the entire study can be entirely replicated with new participants and new neural models as well. Indeed, the code supports the huggingface API of generative models.

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

## A  SENSE-MAKING QUESTIONS

In Table 1 we report the sense-making questions used in the study. Note that each pair of code snippet and question has a variable length depending on the programming languages it is written with. The source code snippet can be found here: https://anonymous.4open.science/r/attentionStudyArtifacts-C752/ eye_tracking/source_file_human_dataset_sessions/.

| Snippet Name | Content | LoC | Question |
|---|---|---|---|
| hannoi_Q1 | Tower of Hanoi problem | 28-49 | How does the algorithm moves disks from the starting rod to the ending rod? |
| hannoi_Q2 | | 26-47 | Which is the base case of the algorithm? |
| hannoi_Q3 | | 28-50 | Which is the name of the auxiliary rod in the call TowerOfHanoi(n, 'Mark', 'Mat', 'Luke')? |
| multithread_Q1 | Consumer-producer threads | 106-116 | Is it possible that consumer and producers threads end up in a deadlock state, namely they both wait for each other to finish, but none of them is doing anything? |
| multithread_Q2 | | 104-112 | Is there any line of code in the consumer or producer code which will never be executed? If yes, report it below. |
| multithread_Q3 | | 104-113 | Will the queue object ever raise an exception in this program? If yes, which condition(s) should be met for the exception to be raised? |
| nqueens_Q1 | N queens problem | 78-100 | What does 'solveNQ(-13)' return? |
| nqueens_Q2 | | 78-101 | What are valid dimensions and values for the array 'board'? |
| nqueens_Q3 | | 78-100 | How would you expect the run time of 'solveNQ(n)' to scale with 'n'? |
| tree_Q1 | Recursive tree construction | 87-99 | How many calls to 'constructTreeUtil' will 'constructTree([1, 2, 3], [ 1, 2, 3], 2)' make? |
| tree_Q2 | | 87-99 | Under which conditions could the check 'if i ¡= h' in 'constructTreeUtil' be false? |
| tree_Q3 | | 89-101 | A part of the code you don't have direct access to has called 'constructTree' with unknown parameters. What can you find out about those parameters? |
| triangle_Q1 | Triangle classification | 66-112 | Which of the functions have side effects (namely it modifies some state variable value outside its local environment? |
| triangle_Q2 | | 66-113 | Questions: Which output will you get for the three points [1, 2], [1, 3], and [1, 4]? |
| triangle_Q3 | | 66-112 | What could happen if the call to 'order()' were omitted from 'classifyTriangle'? |

Table 1: Code snippets and related question for each sense-making task.

## B  DETAILS: EXTENDED RELATED WORK

Here we give a more comprehensive description of the related work.

### B.1 Attention on Natural Language Models

Previous work (Jain & Wallace, 2019) on studying attention weights of recurrent neural network models has found that the attention weights do not agree with other gradient based explanation methods and that alternative weights can be adversarially constructed while still preserving the same model prediction. Following work from Wiegreffe & Pinter (2019) showed how the alternative attention weights can be constructed only per a single instance prediction, whereas obtaining a model which is consistently wrong in its explanations is very unlikely to happen.

On the same line, Tutek & Šnajder (2020) has proposed four regularization methods to mitigate the adversarial exploitation of attention weights for recurrent models, resulting in attention weights that when changed are more likely to produce change in model prediction behavior. Among the proposed techniques, the authors mention the use of residual connections, between token embeddings and output hidden state, and a masked language model objective. Notably, both techniques are naturally used by BERT (Devlin et al., 2019), a model based solely on transformers.

This observation agrees with the techniques proposed by Abnar & Zuidema (2020) as postprocessing for attention weights. The authors explicitly account for residual connections when interpreting attention of transformer-based models.

Vig & Belinkov (2019) have studied the attention of a GPT-2 model and observed that some attention heads are specialized in paying attention to specific part-of-speech tags. However, they focus on the smallest version of the model and they only consider a natural language setting of Wikipedia articles.

Multiple previous works Sood et al. (2020); Bensemann et al. (2022); Eberle et al. (2022) have compared model and human attention on several natural language understanding tasks showing how the two are aligned. In particular, on reading comprehension tasks, Sood et al. (2020) show that a more accurate model does not necessarily have a stronger correlation with the humans.

### B.2 Attention Study on Models of Code

Wan et al. (2022) have studied the self-attention of pre-trained models of code to see how the the attention of those model can be used to induce the abstract syntax tree of the source code. However they limit to relatively small transformer models for code understanding which use only self-attention, whereas we study the first to study the largest version of CodeGen Nijkamp et al. (2022), a generative model which pays attention only to preceding tokens. In the same vein, Zhang et al. (2022) have shown how the attention weights of pre-trained model on source code capture important properties of the abstract syntax tree of the program pointing towards the idea that the attention weights of pre-trained model on source code, which we now disregard, might contain valuable information.

### B.3 Eye-Tracking Studies

An eye-tracking study involving 38 students fixing or describing five simple Python and C++ programs (5-13 LoC) has shown that the fixation duration is comparable between the two languages (Turner et al., 2014). Beelders (2022) qualitatively observe the eye movement of 36 students and four lecturers when reading and mentally executing a short C# program (12 LoC). An eye tracking dataset with 216 participants was collected by Bednarik et al. (2020), however they only consider two short snippets (11-22 LoC) of code, since they do not support scrolling. Blascheck & Sharif (2019) and Busjahn et al. (2015) instead studied the reading order in C++ and Java code comprehension task focusing on six small programs that could fit into a single screen, whereas we consider longer snippets and a much larger dataset of 45 unique tasks. Sharafi et al. (2022) have recently studied code navigation strategies on Java code with eye tracking involving 36 participants focussing on the bug fixing process, whose specific goal and might elicit a different kind of reasoning compared to our sense-making tasks.

### B.4 Neural Models vs Humans

Paltenghi & Pradel (2021) have compared the attention weights of particular neural models of code and developers' visual attention when performing a code summarization task, spotting a positive

correlation between the two. They found a stronger correlation on the *copy attention* mechanism, which an instance of a pointer network (Vinyals et al., 2015) used by the model to copy verbatim the code from the input to the output. However, they use a deblurring interface which might bias the cognitive process of the developer, wehreas we use eye tracking data which are collected in the more natural IDE setting.

### B.5 ALTERNATIVES TO ATTENTION

There are other observable signals than attention that might capture the concept of relevance. In particular, there have been some approaches based, partially or entirely, on gradient information, such as GradCam Yuan et al. (2021); Chefer et al. (2021). These are promising avenues, however, we focus on attention for two reasons: (1) almost all state-of-the-art models of code are based on the transformer block, and the attention mechanism is ultimately its fundamental component (Vaswani et al., 2017), for which reason we expect the corresponding attention weights to carry directly meaningful information about the decision process of these transformer-based models; (2) unlike the gradients, the attention weights can be extracted almost for free during generation with little runtime overhead, a crucial condition in many practical scenarios.

### B.6 IMPORTANCE OF CODE NAVIGATION

A recent study by Peitek et al. (2022) has found that a higher program comprehension efficacy is correlated to a lower coverage of code elements, namely developers with a better code comprehension also navigate the code with a more focused approach. Their results shows how an accurate code navigation is crucial for the efficacy of code comprehension and further motivates our work. Minelli et al. (2014) studies and visualizes how developers interact with a real IDE, however they only use the events of the IDE without any eye tracking date, thus the granularity of our dataset is higher.

## C   DETAILS: ANNOTATION PROCEDURE

We annotate each answer generated by either the developer or the model involving four annotators in the process. We use a scale of three values of correctness: (1) *correct*, when the answer touches all the expected correct points, (2) *partial*, if at least part of the correct answer is present or if the answer is wrong but in the same style of the correct solution (e.g. the Big-O notation), (3) *wrong*, when the answer does not contain any correct part of the answer. Note that, especially for the model, if the model generates extra text beyond the correct or partial answer we ignore the rest if it is incorrect. Moreover, whenever the question is under-specified we accept multiple correct answers as long as they are compatible with the question.

To ensure a reproducible annotation process, all the authors collectively come up with a shared set of gold standard answers for each question. Then, two of the authors independently annotates more than 20% of the answers generated by the model and the developer, and within two rounds of annotation followed by discussion, the final set of gold standard answers is agreed upon. The final agreement on the 20% of data led to a Cohen's Kappa of 0.898 and 0.711, which is considered a very high agreement McHugh (2012). Finally, the remaining 80% of the data is split in half and annotated only by one of the two authors individually.

## D   DETAILS: EYE TRACKING DATA POST-PROCESSING

We use an eye tracker from GazePoint, which is placed below the monitor thus not requiring the user to wear any additional device. The participants can see between 21 to 26 lines of code. The screen size is 52.7 mm x 29.6 mm with a resolution of 1920x1080 pixels. The participant seats at a fixed distance of ca 30cm from the screen.. We do not enforce this via a resting support for the chin, since the eye tracker specification claims to be robust to this kind of movements. We collect eye fixation events $evt_{eye}$ as a tuple $(t, x, y, d)$ where $t$ is the timestamp in milliseconds, $x$ and $y$ are the coordinates of the fixation point in pixels and $d$ is the duration of the fixation in milliseconds.

Beside the eye data, we also continuously log the visible text in VSCode by using a custom plugin, so that we can know what was visible at a given x, y coordinate at each point in time. A visible text

event $evt_{txt}$ corresponds to a tuple $(t, txt, f, l)$ where $t$ is the timestamp in milliseconds, $txt$ is the visible text, $f$ is the file name shown in the code area, $l$ is the line number of the first visible line with respect to the given file. Note that this data source is crucial since we study long code snippets and allow also screen scrolling. To ensure that we have a consistent grid mapping between pixel positions and char positions in the text, we use monospace font and prevent partial scrolling (i.e. where only half a line is scrolled). Thus via our plugin we allow only scrolling of a discrete number of lines.

For additional sanity check we also collect a screen recording along with the eye tracking recording. At any point in time the video shows where is the fixation of the participant.

Note that our setup is as close as possible to a normal coding session, without any invasive method.

To derive the developer attention maps from the eye tracking data, we first synchronize the two data sources, i.e. from the VSCode plugin and the eye tracker, via their timestamps.

Then we derive the x and y coordinate of the code area displayed on the screen for a given video frame by using two visual markers, which delimit the top-left and bottom-right corners respectively. Note that, since the plugin prevents any resizing of the code area during the experiment, any frame would be suitable, for our case we use the frame in the middle of the video screen recording. Then, knowing the fixed number of lines and columns displayed in the code area, we divide the code area in small rectangular areas, each of which corresponds to a specific character position identified by a line and column.

Then, we convert the fixation point of gaze $x$ and $y$ coordinates of each $evt_{eye}$ to the corresponding character position in the code area, in the form of line and column coordinates in the relative coordinate system of the code area. And knowing the line number $l$ we can attribute the fixation to a specific character position in the original file. In this way, we convert each $evt_{eye}$ its equivalent event in character coordinates $evt_{eye}^{(char)} = (t, c, l, d)$ where $t$ is the timestamp, $c$ and $l$ are the column and line coordinates with respect to the original file of the fixation point and $d$ is the duration of the fixation.

Since it is hard to tell whether a participant is looking at a specific character or a group of neighboring characters during a single fixation points, we attribute the attention of the developer also to neighboring characters, namely, if the developer looks at position $(c, l)$ in the original file, we augment our data by introducing new events which point to all the neighboring characters within a vertical offset $v_{off}$ and an horizontal offset $h_{off}$ from our coordinate $(c, l)$. As a result, we replace each basic $evt_{eye}^{(char)} = (t, c, l, d)$ with the set of derived events $(t, c_{new}, l_{new}, d)$ where $c - h_{off} \leq c_{new} \leq c + h_{off}$ and $l - v_{off} \leq l_{new} \leq l + v_{off}$.

Indeed, as reported by Schotter et al. (2012) our fovea region, which is responsible for a sharp central vision, accounts for 2° of the visual field, whereas the parafoveal region, which is used for visual search and scene perception, accounts for 5° of the visual field. Thus considering 5° visual region and our screen size (527mm x 296mm) the developer can see 7.16 characters horizontally and 2.92 characters vertically. Rounding those quantities we set $v_{off} = 1$ and $h_{off} = 4$. This approach also contributes to mitigate eventual small $x$ and $y$ errors in the eye tracking data collection.

Note that that our offset allows to give attention also to blank spaces which are not present in the original line (e.g. after the end of line), thus we do not assign any weight to those characters for a better comparison with the neural models which cannot see them.

# E   EYE TRACKING DATASET

The collected eye tracking dataset contains data from 25 participants over 92 valid sessions. We discarded the data of the sessions where two participants who added comments in the code since it was unclear how to treat the fixations on this newly created text; especially when computing the connection strength between two interaction events with an interleaving fixation event on this newly inserted comment. Figure 8 shows the number of participants per task and programming language.

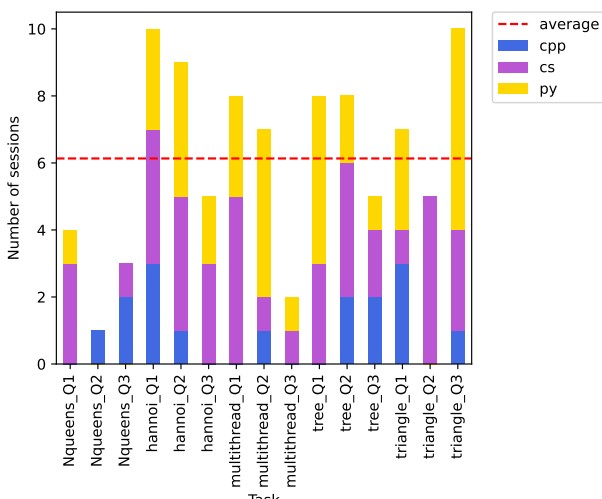

Figure 8: Total number of participants per task divided by programming languages.

## F  OBSERVED DEVELOPER CODE TRAVERSAL

Developers do not jump randomly between likely locations within the code: they have a significant bias for staying close to their current position. We therefore attempt to predict the observed strength of the temporal connection between tokens solely on the basis of their relative position.

We suggest a two-tiered approach: first consider whether the developer is traversing forwards or backwards, then use a relative model for *how far* they will move in that direction.

We expect the ratio of forwards or backwards traversal to be dependent on the exact task, and in fact in our data set its average by task ranged from `5:6 (0.846)` to `7:2 (3.58)`. For each task individually, however, as well as to some extent in general, the best simple predicting feature appears to be the current token position divided by the total number of tokens, i.e. the ratio of the document still in front of the developer. Fitting individual linear regressions for going forward and backward (which do not sum up to 1 because of the chance of returning to the token itself) gives the predictions of

$$\sum_{j>i} S_{i,j} \approx 0.940639 - 0.745988 * i/max(i) \ (R^2 = 0.56) \tag{3}$$

$$\sum_{j<i} S_{i,j} \approx 0.052612 + 0.745747 * i/max(i) \ (R^2 = 0.56) \tag{4}$$

We expect, and find, the distribution for the distance to depend less on the task. Of a number of standard distributions we tested against (normal, poisson, lognormal, exponential, Pareto, Weibull), it is by far best modelled using a fitted Weibull distribution, with best fit of `shape = 0.8932339, scale = 98.1442010 tokens` going forward and `shape = 0.8831494, scale = 105.6159039 tokens` going backward (see Figure 3).

## G  TASK CORRECTNESS

Figure 9 shows the correctness of the answers for each task, for the three models and for the developers.

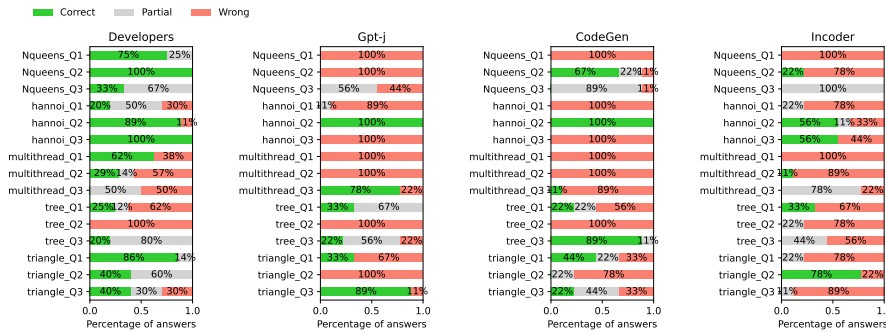

Figure 9: Percentage of correct, wrong and partially correct answer for each tas.

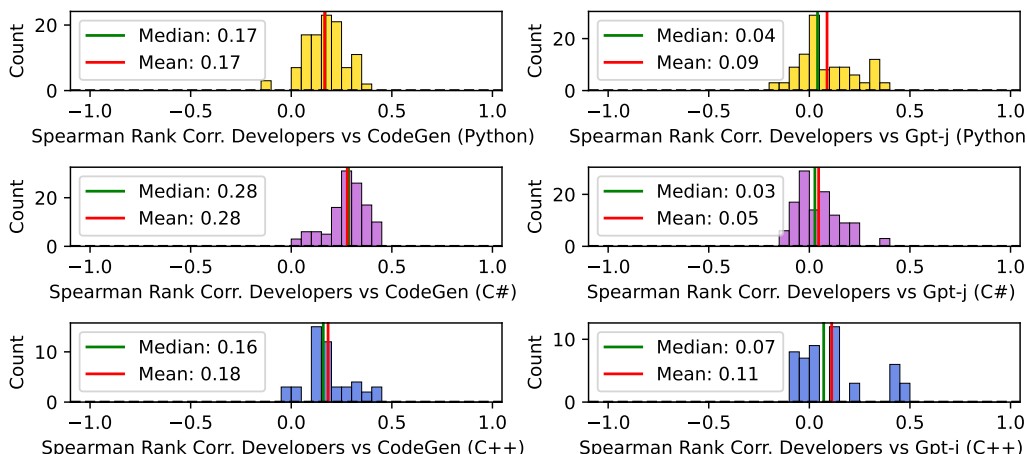

Figure 10: Agreement between developers and CodeGen's visual attention (extraction function: max) while performing code understanding tasks in different programming languages.

## H PROGRAMMING LANGUAGE ANALYSIS

*Is there any difference between programming languages?* To assess the difference between languages, we group the datapoints in three groups, one for each language and we report mean and median for each group in Figure 10 and we compare the distribution of the Spearman rank correlation coefficients via the Mann Whitney U Test (Mann & Whitney, 1947). The test reveals that the agreement between developer and models is significantly different between C# and Python snippets (see test results in Table 2). Although, a visual inspection shows how the two models do not have the same tendency, with CodeGen agreeing more on C# and Gpt-J agreeing more on Python.

In Table 2, we report the results of the Mann Whitney U statistical tests when comparing the distributions of developer-model agreement (Spearman rank coefficients) on tasks written in two different languages.

## I AGREEMENT ON INTERACTION MATRIX: GPT-J AND INCODER

In Figure 11 and 12, we report the results for the agreement on the interaction matrix between human and GPT-J and InCoder-6B.

| Neural Model | Lang. A | Lang. B | Statistic | p-value | Result |
|---|---|---|---|---|---|
| Gpt-j | py | cpp | 2613.0 | 9.37e-01 | Probably same distrib. |
| Codegen | py | cpp | 2499.0 | 7.22e-01 | Probably same distrib. |
| Gpt-j | py | cs | 7637.0 | 2.00e-02 | Probably diff. distrib. |
| Codegen | py | cs | 2626.0 | 9.29e-15 | Probably diff. distrib. |
| Gpt-j | cpp | cs | 3404.0 | 6.61e-02 | Probably same distrib. |
| Codegen | cpp | cs | 1486.0 | 9.95e-07 | Probably diff. distrib. |

Table 2: Results of the Mann Whitney U statistical tests when comparing the distributions of developer-model agreement (Spearman rank coefficients) on tasks in two different languages.

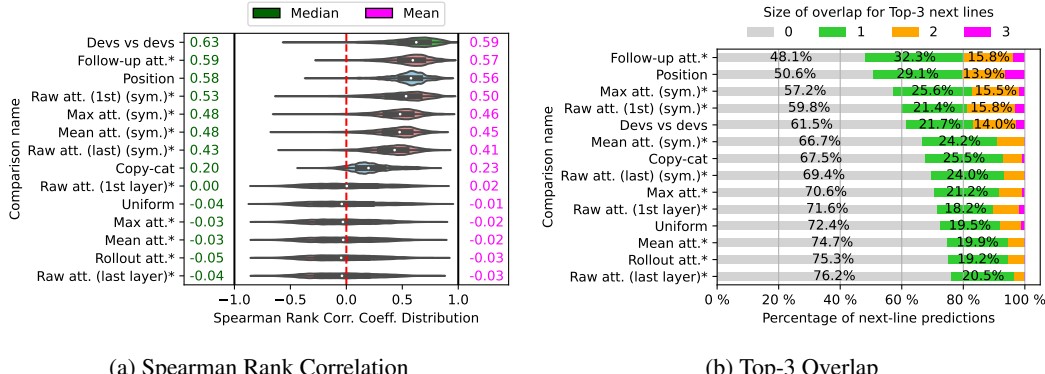

(a) Spearman Rank Correlation      (b) Top-3 Overlap

Figure 11: Agreement between developers and GPT-J interaction matrix ordered from weak (top lines) to strong (bottom lines). Attention-based methods are marked with *.

## J  AGREEMENT AT TOKEN-LEVEL

In Figure 13, we report the agreement computed at the token-level for the three models. Since they have different tokenizers, especially regarding how they treat spaces, we compute the correlation on the tokens that contain some meaning, namely we approximate this concept by considering only tokens with at least one alphabetic character. Indeed, eye tracking data cannot pinpoint with absolute precision when a user was looking at a single space char rather than its surrounding tokens, thus excluding them avoids the result being biased by this irrelevant tokens, from a human perspective.

## K  AGREEMENT VS ACCURACY

In Figure 14, we report the human-vs-model agreement for different pairs of accuracy buckets measured with wrong, partial or wrong. For example, the cell wrong-wrong is contains the correlation when both models and human produce a wrong answer to the same question, whereas the cell correct-correct contains the correlation when the produce two correct answers.

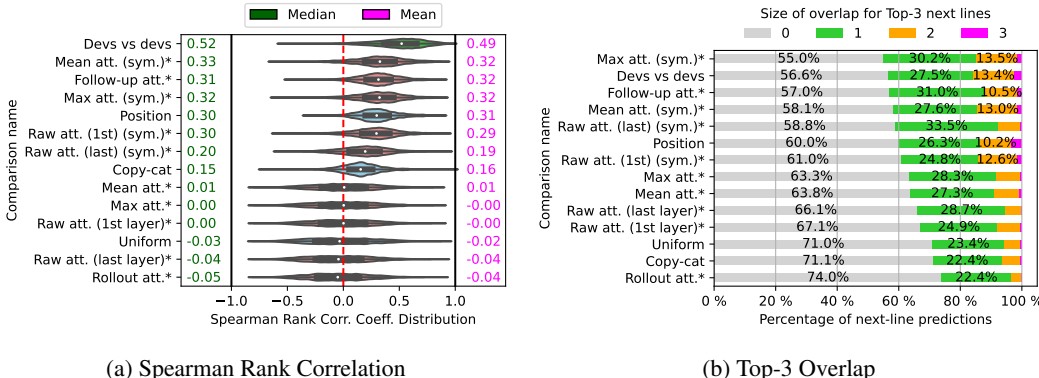

(a) Spearman Rank Correlation

(b) Top-3 Overlap

Figure 12: Agreement between developers and InCoder interaction matrix ordered from weak (top lines) to strong (bottom lines). Attention-based methods are marked with *.

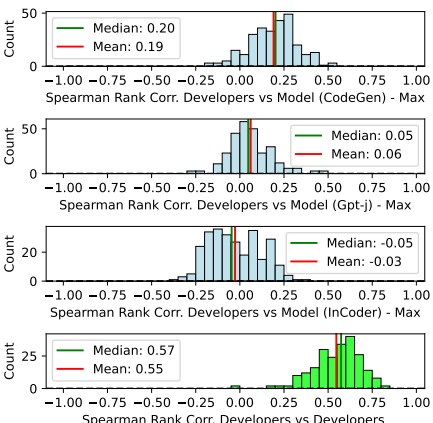

Figure 13: Agreement between developers and models' visual attention (extraction function: max).

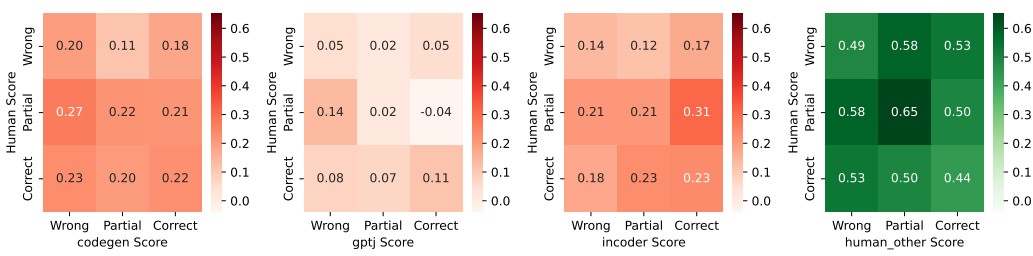

Figure 14: Human-vs-Model agreement for different accuracy pairs.

