# OpenReview forum: "Extracting Meaningful Attention on Source Code: An Empirical Study of Developer and Neural Model Code Exploration"
_ICLR.cc/2023/Conference — Submitted to ICLR 2023_

### Official Review · Reviewer_AZZZ · 2022-10-24

**Confidence:** 3
**Correctness:** 3
**Technical Novelty And Significance:** 3
**Empirical Novelty And Significance:** 3
**Recommendation:** 6

**Clarity, Quality, Novelty And Reproducibility:**

Clarity:
- In Figure 4, in addition to presenting aggregate results for different question types, it would have been nice to see a breakdown at the question-level to make comparisons at a more fine-grained level.
- In Section 3, it is written that some questions were sourced from the Internet and some were written from scratch by the authors. More details are needed about exactly how these questions were collected from the Internet/written by authors.
- Figure 6 is not referenced in the paper, and it is not clear whether GPT-J or CodeGen is the underlying model here.
- On Page 8, "which are those introduced in Section 6.1" seems to be a wrong reference.

Novelty:
- There is some empirical novelty in the newly collected eye-tracking dataset, though eye-tracking datasets have been previously collected for code-related tasks.
- There is technical novelty in the follow-up attention algorithm which combines the temporal dimension with attention layers and some fine-grained analysis is done with this in Figure 7.

Reproducibility:
The authors have provided a link to the data and they use publicly available models, so it would not too difficult to replicate the results.



**Strength And Weaknesses:**

Strengths:
- With growing work in the space of understanding the reasoning capabilities of large language models, I find this to be a very interesting piece of work which tries to relate human reasoning with model reasoning through proxies (eye-tracking and attention).
- The idea of leveraging the temporal relationship through layers to process attention weights is quite novel, and the impact that it has on predicting the next line the developer is going to look at is impressive.
- The authors have carefully controlled for various side effects (e.g., neighboring effect, temporal connect) when running experiments and presenting for results.
- Extensive empirical analysis of attention post-processing techniques, which could be useful for future research which employs attention weights of models for different end tasks.

Weaknesses:
- The underlying motivation for this work is comparing the alignment between developer attention and model attention when solving the task. Since the eye-tracking data is collected while the developer is writing the answer to the question in the prompt, I can see how this can be considered an appropriate proxy for what the developer is looking at "when solving the task." However, my understanding is that the model's attention weights are gathered from the teacher-forced prompt tokens only, and not from when new tokens are being generated to actually solve the task. Therefore, I question whether this is in fact an appropriate proxy.
- From Figure 4, the performance of the models are quite low, with them only achieving correctness 23.7% of the time. This suggests that these models are not performing well on the sense-making tasks. Since they are not predicting correct answers, isn't it possible that their reasoning that led to these incorrect answers is also incorrect? So, does it really make sense to analyze and make claims based on these attention weights?
- In Figure 4, CodeGen and GPT-J achieve very similar performance. However, in Figure 5, their correlation with developers is very different, with GPT-J achieving very low correlation. This seems to suggest that models may attend to the input context very differently from one another and still achieve similar results. Additionally, the developers vs developers correlation is also only moderately positive, suggesting that developers may inspect the input context differently and still provide the similar answers to the sense-making questions. Therefore, it seems like there is a lot of variability in proxies like eye-tracking and attention which make it difficult to compare reasoning of models and humans.
- While authors compare to a number of attention-agnostic baselines like cpy-cat, uniform, and position in Figure 6b, I feel that a simple rule-based baseline which always just predicts the next token/next line (which mimics a left-to-right reading paradigm) is missing.

**Summary Of The Paper:**

This paper presents a study which compares developers' attention to input context for sense-making tasks (e.g., mental code execution, side effects detection, algorithmic complexity, deadlock detection) with that of large autoregressive pretrained language models (CodeGen, GPT-J). For this, they collect a dataset through eye-tracking experiments involving 25 developers. For the model, they encode the prompt and analyze how tokens in the prompt attend to one another, across heads and layers. They consider many different post-processing techniques for attention, including mean, max, rollout, and a newly developed technique called follow-up attention that is designed to explicitly model the temporal relationship between attention weights. With respect to visual attention at the character-level, there is a weak positive correlation between model attention weights and developers' attention. For interaction attention at the line-level (given a position in the prompt, which part of the prompt is likely to be attended to next), there is a more moderate positive correlation. They consider the task of predicting the next line that the developer is likely to look at next, and using follow-up attention yields the best performance for this.

Contributions:
- A large eye-tracking dataset corresponding to different sense-making code tasks
- Empirical results for how the attention employed by models compare to developers
- The follow-up attention processing technique which provides more signal on what line a developer is likely to look at next, which could be useful for supporting end tasks related to code navigation


**Summary Of The Review:**

Overall, the study presented in this work draws some interesting connections between how developers inspect code and the attention weights of large language models. The authors also present a novel algorithm for post-processing attention which incorporates the temporal aspect. However, I have some concerns about whether the attention weights which were extracted actually align with how a model is solving the underlying task and whether the low model performance suggests that it might not be attending to appropriate tokens. Additionally, it seems like there is a lot of variability in eye-tracking data and attention weights. So, I find some of the comparisons and claims made in this paper to be not as convincing.

---

> ### Author Response · Authors · 2022-11-15
> **Response to Reviewer AZZZ**
>
> Thanks for the kind words regarding our work in terms of rigor and novelty! We also appreciate the very precise suggestions to improve the clarity of the presentation.
> 1. Regarding the way in which the attention weights are gathered, for all our experiments we consider the attention paid by both prompt and generated tokens, since as you said the generated tokens carry important task-specific information. Thanks for the question, we will make sure to add an extra sentence to state this clearly in the final revision. This concept indeed is only implicitly mentioned in our mathematical formulation, where tensor A in equation 1 has shape (l, h, n + m, n + m), where n and m are respectively the number of prompt tokens and number of generated tokens, and in the fact that the extraction functions g take the entire tensor as input.
> 2. Regarding the effectiveness of the model under study, we acknowledge that the models perform worse than humans on average. But these questions are also genuinely difficult including run-time analysis and multithreading topics (see Appendix A). Thus, on the other side, we are surprised that they are still able to achieve non-trivial performances in a zero-shot scenario like this, where the model was never trained on this specific format.
> Even if the models do not show developer-like performance yet, we still believe there is value in comparing them for two reasons: (1) to better understand what is missing to the models to reach that point, (2) sometimes, when given hard questions, despite having looked at the right code, even developers could give wrong answers, but we cannot conclude that the reasoning was completely wrong. In the same way, we rely on the assumption that even if the answers of the models are not correct, there might still be value in the reasoning behind it, similarly to what a human sometimes does. After all, most individual reasoning steps may be correct even if the final answer is not.
> 3. Regarding the variability in our eye-tracking data, we agree that it is challenging to compare model reasoning and human data when it is possible that two humans solve the same task but following two different paths. For example, we observed that in two questions, where the question was about the parameters of a function, one participant looked at its definition, whereas the other looked at its usage in the code. Although we cannot prevent this from happening, we interpret the high developer-developer correlation in figure 5 as a good clue that there is an overall ground truth. Moreover, the +0.51 average we found is close to the +0.59 found by Paltenghi and Pradel, 2021 and the difference could be explained by the fact they consider snippets which are much smaller (11.9 LoC, whereas for us 72.2 LoC). To mitigate this variability we always draw conclusions only based on the average behavior, avoiding the discussion of specific cases.
> 4. Regarding the suggestion of a simple next-line baseline, our position-based baseline is rather similar: iit predicts the neighboring line and previous line (and, failing that, the lines of distance 2, and so on). In particular, if the “always-predict-next-line” rule, as we understand it, has a positive top-3 overlap, then “position-based” should as well. Thus, we believe the baseline “position” should be enough to study this aspect.

---

### Official Review · Reviewer_oGLs · 2022-10-26

**Confidence:** 4
**Correctness:** 2
**Technical Novelty And Significance:** 2
**Empirical Novelty And Significance:** 2
**Recommendation:** 3

**Clarity, Quality, Novelty And Reproducibility:**

The presentation quality is good. It is not clear if the experiments are repeatable in this nascent area as the experiments have lot of moving parts (task, expertise of subjects, equipment used, etc.). The same experiment in a different organization can lead to somewhat different results.

This aspect is not used to penalize the paper in any sense.


**Strength And Weaknesses:**

(+) Correlating model’s attention with human attention is an interesting area of research. The eye tracking setup used in the paper is more natural compared to other studies.

(+) The paper proposes “follow-up attention” which is marginally better.

(-) The paper needs to discuss how even attention-agnostic baselines perform well.

(-) Figures 6 and 7 need to be reconciled. Raw attention in layer 1 performs poorly (Figure 6) while follow-up attention with only layer 0-1 is competitive.  The reasons need to be discussed.

(-) In Figure 6(a), it might appear higher top-3 overlap should indicate better performance. Follow-up attention has lowest score. Explain.


**Summary Of The Paper:**

The paper uses eye tracking studies on developers and correlates the patterns with the attention from code-related deep learning models (CodeGen and Code-J). CodeGen has higher agreement with developers (median = 0.23). The paper also reports interaction patterns. While attention-agnostic schemes (such as position) do well, the proposed “follow-up attention” has the best agreement.

**Summary Of The Review:**

The paper addresses an important topic of correlating human attention with model attention. The authors have done a commendable job of using non-attention based baselines and reporting the performance. A deeper interpretation of the reported results, especially the good performance of non-attention baselines, is required to benefit other researchers.

---

> ### Author Response · Authors · 2022-11-15
> **Response to Reviewer oGLs**
>
> Thanks for the appreciation regarding the presentation quality; and we are glad that you also agree with us on the importance of correlating human attention with model attention.
> We also appreciate your concerns, and address them as follows.
>
>
> 1. The best performing agnostic baseline is the position-based, and its relatively good agreement is definitely interesting to further discuss. This surprised us as well at first, but it can be at least partially explained by the developers’ reading style being still influenced by the way in which we read natural language. Indeed, previous work (“Eye Movements in Code Reading: Relaxing the Linear Order”, 2015 IEEE 23rd International Conference on Program Comprehension) showed that programmers still exhibit a strong linear scanning of the source code 60-70% of the time, as compared to 80% measured on natural language. Even if lower than for natural language, this effect is still present in source code, and the good performance of position-based baseline probably reflects this.
> Regarding the other baseline copy-cat, it is clearly worse than the attention-based counterpart, showing how a naive similarity approach is not good in predicting user interactions.
> From a practical application viewpoint of course, the position based component is less interesting that the others, since developers don’t need any help with proceeding linearly. Whereas knowing which line a language model would give attention to next could be a better candidate to support code exploration in a more complementary way.
> We plan to incorporate this discussion of the agnostic baselines in the next revision, and we are confident it will significantly improve the paper discussion. Thanks.
>
>
> 2. We appreciate our presentation might be insufficiently clear in this respect and have tweaked it in our revision. Please comment further if it is still not sufficient. The figure 6 “raw attention 1st layer” refers to the use of the attention of the first layer which is a triangular matrix (each token can only attend the preceding ones). As you said, it performs poorly, as does raw attention of the last layer, or attention aggregated over layers in the most direct ways (mean or max).
> Whereas Figure 7, when saying Layer 0-1, refers to our formulation of the follow-up attention described in the algorithm, but considering z = 1 only. Thus layer 0-1 means computing the follow-up attention considering the first layer and the second layer (z+1). A characteristic of the follow-up attention is that it starts from triangular raw attention matrices and creates a matrix which is fully dense. The ground truth interaction matrix is computed as a dense matrix. Thus at least part of the difference between the “raw att. 1st layer” and “follow-up layers 0-1” can be explained by the fact that the first compares a triangular matrix to the ground truth (dense), whereas the second compares a dense matrix to a dense matrix leading to better results.
> Another complementary and more direct explanation could also be that the novel computation of the follow-up attention is responsible for the better performance than the raw attention, further motivating our work.
>
>
> 3. Regarding Figure 6, we’d suggest this question might have been triggered by a misunderstanding, due to the admittedly unusual way of showing the best method at the end of the figure. Indeed, the follow-up attention obtains the highest (not lowest) score for the top-3 overlap, and the second highest for raw correlation, beaten only by “result of another developer”. In case the ordering was the cause of the misunderstanding, we have changed the reviewed version to show the best method on top. Please reply if this is not the actual reason, or if you consider the remedy not sufficient.

---

### Official Review · Reviewer_6UaZ · 2022-10-27

**Confidence:** 4
**Correctness:** 3
**Technical Novelty And Significance:** 2
**Empirical Novelty And Significance:** 2
**Recommendation:** 5

**Clarity, Quality, Novelty And Reproducibility:**

# Clarity & Quality
The paper was generally well-written, and the results were clear and presented well. There are grammatical mistakes throughout, but they don't significantly impede understanding. There are also several small mistakes (e.g. "which are those introduced in Section 6.1" on page 8 should probably refer to 5.2 I imagine? And "Figure ??" at the bottom of page 18 should probably be corrected), but generally the paper is clear.

# Novelty
The three main points of novelty in this paper are:
- An eye-tracking dataset that differs from previous one in terms of the lengths of programs and the tasks examined (though it's not clear to me why this distinction is significant for the particular problem examined in this paper).
- Framing the examination of attention in terms of code exploration (though since no impact of this is measured and the results are not compelling, it's hard to know how significant this is).
- Follow-up attention as an extraction function to make an interaction matrix (though since this isn't benchmarked on an existing task, it's hard to tell how this performs in practice).

# Reproducibility
Since the code and dataset have been released, the paper should be reproducible, though I haven't verified this.

**Strength And Weaknesses:**

# Strengths:
- The paper is clear and well-written, explaining everything necessary to understand the paper.
- This is an interesting approach that could lead to a useful developer tool based on helping people understand code faster.
- The code and dataset were released, which is useful for follow-up work.


# Weaknesses:
- The results seem not that impressive.
  - The authors defined their own methodology and metrics, and then reported results on them, which makes it hard to understand the significance of the results (in particular, the follow-up attention method introduced in this paper).
  - For the interaction matrix results, the attention-agnostic code retrieval patterns seem to perform as well or better than the attention mechanisms?
  - There are no baselines for visual attention, so it's hard to extract meaning from the values given (beyond that the correlation is positive). Also, based on the results on the interaction matrix, I would guess that attention-agnostic predictions could also outperform attention based ones.
  - There are no real measures of usefulness, or investigation into how this could be useful. Based on the results presented in this paper, it's hard to judge how useful a next step would be.
- As part of the results seeming not impressive, it's hard to judge how significant or impactful this work is. It proposes an idea for how to use attention to create a developer tool, but not a compelling case of why attention is better than relatively simple alternatives, or why this tool is useful.

Some ways of improving this would be to see if the predictions produced by this model are helpful for developers in practice, split the dataset into train/test/validation and train some baselines on this (e.g. freeze a transformer and train an additional layer on top of the last attention block to predict visual attention), or add some basic baselines to visual attention (e.g. favor earlier tokens, or favor later tokens, or generally look at the developer patterns and try to come up with simple rules to explain them).

**Summary Of The Paper:**

This paper focuses on attention activations in conventional transformer models: what they learn, and what information can be extracted out of them for other use-cases. Models are usually trained directly for another objective (e.g. completion), but the attentions themselves can be used with no or minimal processing for other tasks, because they capture important features of the input data. Attentions have been shown in previous work to be useful for various tasks, including saliency and interpretability.

The authors focused on code models, and in particular tracking what developers pay attention to when looking at code. They collected/released a dataset that tracked what programmers looked at when solving various problems, and compared this to what several different transformations (including one they introduced) of the attention activations would have predicted. They compared both how often each token was perceived and what token was looked at after each token, and found that attention can be useful for predicting these. Using attention in this way has promising implications for developer tools, such as suggesting which lines developers should look at when exploring an unfamiliar file.

**Summary Of The Review:**

This paper proposes using attention for code exploration, and examines how well attention performs at predicting how developers explore code. The paper as a whole is well-written and presents its results clearly, but based on the evidence presented the significance of the work is somewhat questionable. There is no strong evidence presented of how this could lead to a useful development tool, or that attention activations are better for code exploration than relatively simple baselines.

---

> ### Author Response · Authors · 2022-11-15
> **Response to Reviewer 6UaZ**
>
> Thanks a lot for the positive feedback on the clarity of presentation. We are also glad that you raised the discussion on what comes next in terms of practical tools.
>
>
> Focusing specifically on your concerns, in order:
>
> 1. Regarding the significance of results, we want to highlight how the goal of this work is to contribute a first approach using the attention signal for a direct practical usage other than code generation, for which it was designed. By performing the study and evaluation of our attention-based post-processing baseline we have also developed the follow-up attention as a possible sensible processing scheme. But while we are glad it performed well, the aim of this work is more general than pushing this concept of follow-up attention only. Rather, we want to establish this new task as something challenging and valuable on which we now have provided a reusable dataset.
>
>
> 2. Regarding the motivation on the methodology and metrics used, we tried to build on existing literature whenever possible. For visual attention, we borrowed from previous work for the correlation study (Paltenghi & Pradel, 2021); whereas for the evaluation of the attention-based methods when predicting the next-line, we use adapt a common strategy used in Recommender System literature using offline data, namely checking whether the predictions conform to what was done in the past by the developers (aka our eye-tracking dataset). There is the alternative to either gather a global indicator of the match between the two rankings, for which spearman rank is the most established method. Alternatively, it is of particular interest to report on the local agreement on the most important items, which led us to the top-3 overlap as a simple and straightforward measure.
>
>
> 3. Regarding the performance of the agnostic baselines, we redirect you to the answer given to the reviewer oGLs point 1, where we explain more in detail how the good performance of the position-based baseline (the only agnostic one which is comparable with the attention-based one) could be the result of the linear reading order still present also when reading code, even if less evident than with natural language. (“Eye Movements in Code Reading: Relaxing the Linear Order”, 2015 IEEE 23rd International Conference on Program Comprehension)
>
>
> 4. Regarding the usefulness of a next step and the unsuitability of the agnostic baselines, we stressed in the previous point how position-based is indeed useless when implementing a practical tool which recommends where to look at next, since developers already know which line is next. Alternatively, looking blindly to the other occurrences of the same token in the text, as the copy-cat does, does not significantly improve on the IDE search function. It turns out not to be a very good indicator for human code navigation in any case, and for example when there are multiple occurrences of the same variable in the text; the copy-cat has no knowledge and no way to tell which occurrences are most likely linked.
>
>
> 5. Regarding a further user study to assess the usefulness of a potential tool, our work primarily wants to provide (1) the foundations of the problem, (2) a reusable eye-tracking dataset, (3) a first evaluation of ten approaches, rather than push a specific tool or algorithm as “the” only solution. Nevertheless, we definitely agree on the value of such a study and we definitely endorse it as interesting future work, especially for a more tool oriented piece of work.

---

### Official Review · Reviewer_ZiWC · 2022-10-28

**Confidence:** 3
**Correctness:** 4
**Technical Novelty And Significance:** 2
**Empirical Novelty And Significance:** 3
**Recommendation:** 6

**Clarity, Quality, Novelty And Reproducibility:**

Clarity:

The paper is well written

Quality:

It would be good to have p-values for the correlation coefficients for the sake of completeness. That said, visually looking at the distributions, the results do seem statistically significant

Novelty:

The eye tracking dataset is novel. The new attention metric of followup attention is novel as well.

**Strength And Weaknesses:**

Strengths:

(1) The method of eye tracking is well conceived and executed

(2) There is a thorough exploration of various attention metrics and their relation to eye-tracking

(3) Care is taken to properly define what interaction means in context of eye tracking -- as users can casually skim through code without attention to one piece of code necessarily being prompted after reading another piece of code

(4) Follow-up attention is a novel metric that shows much improved correlation with eye tracking data

Weaknesses;

(1) Only CodeGen and GPT-J (which is not a code generation oriented model) are tested. It would be better to test on other models like Incoder and others that have comparable performance to CodeGen and see if the findings can be replicated.

(2) Do we see higher attention matrix correlation when both the model and the developer have the right answers vs. when they don't?

(3) It would be nice to see how much the attention matrix is correlated with the data flow or call graph -- and if edges between tokens in the interaction matrix that correspond to data flow are removed -- whether we still obtain a correlation between the attention matrix and the eye tracking dataset. This would help understand what aspects of the code induce the Transformer attention matrix to have similarity to the human attention matrix.

**Summary Of The Paper:**

The paper tries to measure the correlation between attention weight patterns for CodeGen and GPT-J on code understanding questions and compare that with eye tracking based attention patterns for humans when answering those same questions.

They use 4 different attention weight metrics --

(1) Attention mean across all layers and heads
(2) Attention max across all layers and heads
(3) Rollout attention
(4) Follow-up attention - a novel metric where they compute what they call follower score which models how the attention flows between tokens across layers. The hypothesis is that this can be similar to how humans focus on one set of tokens and then decide to focus on another set where the second set depends on the first set

They evaluate two different views --

(1) Visual attention -- which is the aggregate attention across time
(2) Interaction matrix -- which is the chance of looking at token i after token j

They find non-trivial correlation coefficients between the different types of attentions and attention metrics derived from humans for visual attention (for mean and max). They also find high correlation as well as predictive power as to what line will be attended next for interaction matrix view -- especially for follow up attention

The correlation is generally a lot less for GPT-J vs CodeGen

They also measure correlation difference between languages and find higher agreement for C# vs Python

**Summary Of The Review:**

The paper provides a new eye tracking dataset as well as a thorough analysis of how it relates to attention in code generation models. Multiple aspects of the attention matrix are evaluated and a novel Transformer attention matrix based metric is developed which shows higher correlation with human attention.

---

> ### Author Response · Authors · 2022-11-15
> **Response to Reviewer ZiWC**
>
> Thank you for the positive feedback, we did indeed put a lot of care in the definition of the interaction matrix calculation from eye tracking data, we are very happy to see this appreciated.
>
> Addressing your remaining concerns one by one:
> 1. Regarding the models choice, we initially picked one model from the NL domain and one from code, but given your comment we extended our dataset to also include InCoder predictions and attention (see paper revision: updated Fig 4 and Fig 5 + Appendix: I).
> The results are in the Appendix I of the revision.
> 2. We computed the correlation when both model and developers are right “Correct-Correct” and when they are both wrong “Wrong-Wrong”.
> We obtain the following correlations for the different models:
>
> CodeGen:
> - W-W: 0.198075 (sigma=0.012649, N=67),
> - C-C: 0.223115	 (sigma=0.010550, N=36)
>
> GPT-J
> - W-W: 0.052541 (sigma=0.011929, N=55)
> - C-C: 0.112980 (sigma=0.022509, N=43)
>
> InCoder
> - W-W: 0.137153 (sigma=0.021986, N=67)
> - C-C: 0.234557 (sigma=0.008736, N=35)
>
> The answer is yes we see this tendency, in line with previous work (Paltenghi & Pradel).
>
> 3. Although for time reasons we had to prioritize other additional experiments (e.g. adding another model), we agree that comparing interaction matrix with traditional software concepts such as data-flow and control flow is an interesting direction for future research.
> Note that for our dataset, this experiment would require significant engineering effort since we include three different programming languages that would need to be reconciled.

---

### Official Review · Reviewer_q7Ss · 2022-10-29

**Confidence:** 4
**Correctness:** 3
**Technical Novelty And Significance:** 2
**Empirical Novelty And Significance:** 2
**Recommendation:** 5

**Clarity, Quality, Novelty And Reproducibility:**

### Clarity, Quality:
- Paper is well written and is easy to follow. Experimental design is discussed in detail. While the paper cites the relevant literature in the code analysis domain, I think authors should also cite relevant NLP papers studying correlation between human and model attention.

### Novelty:
- The proposed dataset is novel.
- To the best of my knowledge, the proposed “follow-up attention” approach is also novel.

### Reproducibility:
- Authors plan to release the dataset so results presented in the paper should be reproducible.




**Strength And Weaknesses:**

### Strengths:

- The proposed dataset will be a nice contribution to the community as it can be used for additional code understanding tasks.
- Proposed “follow-up attention” is very intuitive and is shown to have better correlation with human attention than other approaches.

### Weaknesses:

- Authors study only a single code model Codegen and a single generic language model GPT-J. It’s not clear how model accuracy, model size play a role on the correlation between human and model attention. In particular, do we observe better correlation for larger and/or more accurate models ? On reading comprehension tasks, [3] show that a more accurate model does not necessarily have a stronger correlation so it's important to study such correlation for different models.
- I have reservations about the usefulness of this study. Paltenghi & Pradel (2021) have also studied correlation between human and model attention. Further, multiple prior works [1, 2, 3] have shown that human attention is aligned with several natural language understanding tasks. Authors do not discuss how their work is different from the existing works and in what ways it advances our understanding.
- Some of the design choices are not clear. In particular, authors say that unlike  Paltenghi & Pradel (2021) they study attention at char level. Why? I think developers focus on the whole variable name instead of a single char in a variable name so it’s not obvious why char level analysis is better than token level analysis.

References:
- [1] Bensemann, J., Peng, A., Prado, D., Chen, Y., Tan, N., Corballis, P. M., ... & Witbrock, M. J. (2022, May). Eye Gaze and Self-attention: How Humans and Transformers Attend Words in Sentences. In Proceedings of the Workshop on Cognitive Modeling and Computational Linguistics (pp. 75-87).
- [2]  Eberle, O., Brandl, S., Pilot, J., & Søgaard, A. (2022, May). Do Transformer Models Show Similar Attention Patterns to Task-Specific Human Gaze?. In Proceedings of the 60th Annual Meeting of the Association for Computational Linguistics (Volume 1: Long Papers) (pp. 4295-4309).
- [3] Sood, E., Tannert, S., Frassinelli, D., Bulling, A., & Vu, N. T. (2020). Interpreting attention models with human visual attention in machine reading comprehension. arXiv preprint arXiv:2010.06396.
- [4] Paltenghi, M., & Pradel, M. (2021, November). Thinking Like a Developer? Comparing the Attention of Humans with Neural Models of Code. In 2021 36th IEEE/ACM International Conference on Automated Software Engineering (ASE) (pp. 867-879). IEEE.


**Summary Of The Paper:**

This paper presents an eye tracking dataset for code understanding task in three programming languages (Python, C++, C#). The dataset contained 92 visual attention sessions of 25 developers. Using this eye tracking dataset, authors study the correlation between model attention and developer’s attention. Authors empirically evaluate different attention-based post processing approaches of the model’s attention signal against the ground truth of developers exploring code.

**Summary Of The Review:**

While I don't see any issue with the study design and findings, I have reservations about the usability of this work. In particular, I don't think that this work advances our understanding of the relationship between model attention and developer attention in a significant way.

---

> ### Author Response · Authors · 2022-11-15
> **Response to Reviewer q7Ss**
>
> 1. We agree on the benefit of more models, (also considering the suggestion of Reviewer ZiWC), so we added InCoder (see paper revision: updated Fig 4 and Fig 5 + Appendix: I).
> 2. Regarding the unique points of this work compared to existing literature, first we mention in the text that our work is inspired and builds on Paltenghi & Pradel but with some important differences:
>
> - NOVEL MODELS: They study fairly early neural models of code (CNN attention-based model (Allamanis et al. 2016) and small transformer models (Amhad et al. 2020) based on the initial transformer architecture which could only first 150 input tokens, whereas we focus on a more recent decoder-only architecture (GPT-like), which is key in many language model applications (including commercial ones like OpenAI Codex or GitHub Copilot).
>
> - CHALLENGING TASK: Paltenghi & Pradel focus on the code summarization task using a task-specific model, whereas we focus on a broader and more challenging task (code understanding with question and answers). The code snippets we consider are much longer, their average length in lines of code is 11.9 LoC, whereas for us it’s 72.2 LoC ((26+104+78+87+66)/5 see appendix A). This leads to much richer navigation data.
>
> - NATURAL IDE SETTING: They collect their human data via a de-blurring based interface, which might have influenced, at least partially, the cognitive process of the human.
> For example, when reading blurred code, the human might become more unnaturally selective in where to look and where not to look.
> Moreover, part of their participants were recruited via Amazon Mechanical Turk, making the data collection more cost-effective but making it less difficult to control the legitimacy of their code reading patterns (beside a performance-based filtering).
> Whereas, on our side, we run a more controlled eye-tracking study where each session is monitored by a human and we recruit only real developers.
>
> - CODE-SPECIFIC READING PATTERNS: Thanks for pointing out related works [1,2,3] on natural language, not all of which we were aware of. They appear to be very relevant, and we included each of them in our extended related work section in Appendix. At the same time, they all study natural language which is similar to code (“On the naturalness of software”, 2012), but also lacks some features unique to code such as non-linear scanning pattern, so we definitely believe we provide relevant novelty.
> Figure 5 shows how a mostly NL model like GPT-J is much less correlated than a code-specific ones when looking at source code, thus showing that code and natural language might have different reading patterns.
>
> Overall, this work advanced our understanding of the similarity between pre-trained neural models of code and developers when exploring code, featuring few unique characteristics not found in related work: (1) we study recent and successful open source pre-trained models, (2) we observe the developers in a natural setting leading to a higher quality dataset, (3) we study a more varied code understanding task with significantly larger input code, (4) we study code task which sets this work apart from previous work done on natural language only.
>
> 3. Regarding the minimal unit for our visual attention comparison, we focused on char level because it was natural to our ground truth data collection and directly represented e.g. in our eye-tracking accuracy. Analogously Paltenghi & Pradel (2021) studied on the token-level because the ground truth data was collected at that level, i.e., by deblurring entire tokens. Additionally, token comparison is not uniform among the models because they use different tokenizers. For example, GPT-j’s tokenizer counts every successive space as a single token (meaning a good part of the tokens are indentation spaces), while Codegen’s groups them together in a single token.
>
> Nevertheless, we see the value of comparing token-level attention, in particular regarding your comment about variable names. So we replicated the comparison at the token-level for the models (gpt-j, codegen, incoder) considering all the tokens with at least an alphabet character, to reduce the noise generated by tokens made of only spaces or pure syntactic constructs (e.g. parentheses or dots for method calls). The results confirm that CodeGen still  exhibits the highest correlation (see Appendix K).

---

### Author Response · Authors · 2022-11-15
**General Response to Reviewers**

Thank you to all the reviewers for your time and feedback! We are very glad about the positive and constructive reviews appreciating the work’s strength, and, at the same time, pointing out some potential areas of improvement. To address your feedback we have performed the following updates:
1. added a new model (InCoder) as asked by the Reviewer q7Ss and Reviewer ZiWC to see if the findings can generalize to other models of code. (see paper revision: updated Fig 4 and Fig 5 + Appendix: I).
2. computed the agreement at the token-level as asked by Reviewer q7Ss for uniformity to previous work (Paltenghi & Pradel). (Appendix: J)
3. added agreement human-model for different levels of correctness of the answers (Appendix K)
4. increased clarity, corrected errors, and addressed minor concerns throughout the text.
Thanks again for having contributed to improve the work.

---

### Decision · Program_Chairs · 2023-01-20

**Decision:**

Reject

**Justification For Why Not Higher Score:**

The proposed method and the presented results are not surprising, similar work exist in the past.

Usefulness of the approach is not significantly demonstrated.

**Justification For Why Not Lower Score:**

Can't be lower.

**Metareview: Summary, Strengths And Weaknesses:**

This paper presents a study of the attention matrices in large language models for code and developed a few different ways of visualizing the attention and how that correlates with human developer’s eye tracking data.  This line of work is potentially interesting and the new data could be useful for future studies, but at the current state this paper didn’t excite the reviewers enough to pass the bar for acceptance.  Common issues include concerns over the usefulness and significance of the proposed approach, as prior work has studied very similar problems for language models, and for a future tool for developers, there could be simpler alternatives that are also viable.